# Breaking translational symmetry via polymer chain overcrowding in molecular bottlebrush crystallization

Hao Qi[1,3], Xiting Liu[1,3], Daniel M. Henn[2,3], Shan Mei[1], Mark C. Staub[1], Bin Zhao[2✉] & Christopher Y. Li [1✉]

One of the fundamental laws in crystallization is translational symmetry, which accounts for the profound shapes observed in natural mineral crystals and snowflakes. Herein, we report on the spontaneous formation of spherical hollow crystals with broken translational symmetry in crystalline molecular bottlebrush (mBB) polymers. The unique structure is named as mBB crystalsome (mBBC), highlighting its similarity to the classical molecular vesicles. Fluorescence resonance energy transfer (FRET) experiments show that the mBBC formation is driven by local chain overcrowding-induced asymmetric lamella bending, which is further confirmed by correlating crystalsome size with crystallization temperature and mBB's side chain grafting density. Our study unravels a new principle of spontaneous translational symmetry breaking, providing a general route towards designing versatile nanostructures.

[1] Department of Materials Science and Engineering, Drexel University, Philadelphia, PA 19104, USA. [2] Department of Chemistry, University of Tennessee, Knoxville, TN 37996, USA. [3] These authors contributed equally: Hao Qi, Xiting Liu, Daniel M. Henn. ✉email: bzhao@utk.edu; chrisli@drexel.edu

When matter crystallizes, it follows a defined symmetry to grow, forming profound morphologies ranging from snowflakes to quartz. A pivotal principle in crystal growth is that unit cell repeats itself following translational symmetry, through which the unit cell symmetry is manifested in macroscopic crystals. This translational symmetry, however, can be broken under intrinsic or extrinsic constraints, forming a class of shape-symmetry incommensurate crystals[1–4]. For example, matter can crystallize into helical, helicoidal, and scrolled crystals[5–9], whose shapes are incommensurate with the 3D translational symmetry defined in a classical Cartesian coordinate[1]. Detailed reasons for the formation of these shape-symmetry incommensurate crystals are material-specific, while unbalanced stress is believed to be an important reason for symmetry breaking[5]. For example, in polyethylene, unbalanced stress can arise from chain tilting with respect to the lamellar normal, leading to banded spherulites comprised of helicoidal crystals[5,10–12]. Lamellar unbalance can also be induced by different volumes of the folds as proposed in γ phase poly(vinylidene fluoride) (PVDF) and polyamide 66[5,8,12,13]. Triblock copolymers with crystalline middle block and immiscible end blocks can form asymmetric curved crystals and the unbalanced stress is associated with phase separation of two end blocks[14]. In addition, chiral structure can also lead to symmetry breaking upon forming single crystals[6,7,15].

Here, we report spontaneous translational symmetry breaking and the formation of hollow crystalline spheres with controlled openings, upon crystallization in mBB polymers with crystalline side chains. mBBs refer to a class of polymers with side chains grafted on a long polymer backbone with a sufficiently high grafting density[16,17]. This unique architecture accounts for many newly discovered properties including exquisite mechanical property control, tunable surface friction, sophisticated assembly, and molecular shape changing[18–21]. Crystallization of mBBs has been studied; steric crowdedness can facilitate crystal nucleation and retard its growth[22–27]. Single crystal level study of mBB crystallization, however, has not been reported, while the formation of polymer single crystals (PSCs) could provide a molecular marker for better understanding the chain architecture effect on crystallization. Herein we show that, contrary to the 2D flat lamellae in linear poly(ethylene oxide) (PEO) PSCs, mBBs with PEO side chains grow into 3D spherical hollow crystals with broken translational symmetry, which is attributed to local chain overcrowding in mBBs as confirmed by fluorescence resonance energy transfer (FRET) experiments. The unique crystalline structure is named as mBB crystalsome (mBBC), and we demonstrate that the size and opening of the mBBCs can be tuned by crystallization conditions and mBB side chain grafting density.

## Results

**Spherical crystals of molecular bottlebrushes.** The PEO mBBs used in this study were synthesized by a grafting-to method, where alkyne end-functionalized PEO was grafted onto an azide-bearing backbone polymer using the highly efficient copper(I)-catalyzed azide-alkyne cycloaddition click reaction (Supplementary Fig. 1)[21,28–31]. Detailed synthesis and characterization of mBBs are summarized in the supporting information (SI, Supplementary Figs. 2–6)[21]. Fig. 1a, b show the chemical structures of PEO with a 5k Da molar mass and one corresponding mBB. The mBB polymers are abbreviated as $mBB_n$-$PEO_m$-$100\sigma$, where $n$ and $m$ denote the degree of polymerization (DP) of backbone and side chains, respectively, while $\sigma$ is the side chain grafting density, defined as the percentage of the backbone repeat units that are coupled with a PEO side chain. In this study, $n$ was controlled as 800 and 707, $m$ as 114, while $\sigma$ was varied from 0.10, 0.48, 0.75,

0.76 to 0.94. Table 1 summarizes the molecular characteristics of the polymers. To prepare high quality mBB PSCs, self-seeding solution crystallization was employed. Supplementary Fig. 7 is the typical temperature profile used for the crystallization process[32]. The polymer solution was first quenched to a low temperature for crystallization, and then brought to a self-seeding temperature ($T_{ss}$), at which the previously formed crystals were mostly dissolved and only a trace amount of seeds remained for further crystallization at a predetermined crystallization temperature ($T_c$). The crystal seeds that remained at $T_{ss}$ can be confirmed using dynamic light scattering experiments (Supplementary Fig. 8). They provide heterogenous nucleation sites for subsequent crystal growth and the self-seeding effect on crystal growth is summarized in Supplementary Fig. 9. Using this method, nearly monodispersed 2D and 1D crystals were formed in numerous polymers[33–41]. Fig. 1a shows a transmission electron microscopy (TEM) image of a flat 2D 5k PEO PSC while the inset selective area electron diffraction (SAED) pattern confirms the 4-chain monoclinic unit cell of PEO[42]. When $mBB_{800}$-$PEO_{114}$-76 was crystallized at a $T_c$ of 20 °C for 2 h, curved, nearly spherical morphology was observed (Fig. 1c). While the mBB crystal morphologies are different from the flat crystals in Fig. 1a, the SAED in Fig. 1c confirms that the crystal structure remains the same except that the spotty diffractions observed in the flat crystal become arc-shaped (Fig. 1c), which is typical for diffractions from non-flat crystals due to the inevitable lattice splay in curved space[7,13,35,43–45]. This can also be viewed as packing of small crystallites progressively changing lattice orientation as the crystal grows. The spherical shape can be better viewed in Fig. 1d, e using atomic force microscopy (AFM) and scanning electron microscopy (SEM), where micro-faceted edges (red arrows in Fig. 1e) further confirms the crystalline nature.

The nucleation and growth in the self-seeding process was followed by collecting the crystals at different time points (Fig. 1f–h). The crystal grew from a slightly curved 2D crystal to a spherical shell. The sphere diameter is close to a constant (~2.5–2.8 μm) for the crystals at different growth stages, while the shell cap area gradually increases in the first 30 min (Fig. 1i, Supplementary Figs. 10, 11). Further increasing growth time to 2 h (Fig. 1i) did not increase the shell cap area, implying complete consumption of the mBB in 30 min. To explore if closed crystalline shells can be formed, first, we increased the polymer concentration from 0.01 wt.% to 0.02 wt.% to supply more material and observed a rose-like morphology with multiple layers of open shells (Fig. 1j), which suggests overgrowth on the lamella surface due to the higher mBB concentration. We then added the same mBB into a suspension of pre-formed half-shell crystals to avoid the high concentration-associated overgrowth. Fig. 1k shows that while the opening became smaller, this method could not generate completely closed shells either, possibly because that the individual mBB molecules were too large (~100 nm long, Supplementary Fig. 12) to diffuse and align onto the crystal growth front as the opening of the crystal became increasingly smaller. We then introduced linear PEO (5k Da) to the pre-formed, open mBB crystal suspension and confirmed that linear PEO can continuously grow onto the existing crystals, leading to closed spheres (Fig. 1l). Note that the mBBCs in Fig. 1l do not have a perfect spherical shape. This could be because the intrinsic shape of a linear PEO single crystal is not spherical, but flat. Future work will be conducted to further explore the detailed mBB growth habit near the closure of the mBBCs.

The observed mBB crystals are spherical and similar to the recently reported polymer crystalsomes, which are spherical single crystal-like shells formed when linear polymers are confined to crystallize at curved liquid/liquid interface[43,44]. Accordingly, we coined the name mBBCs to describe these

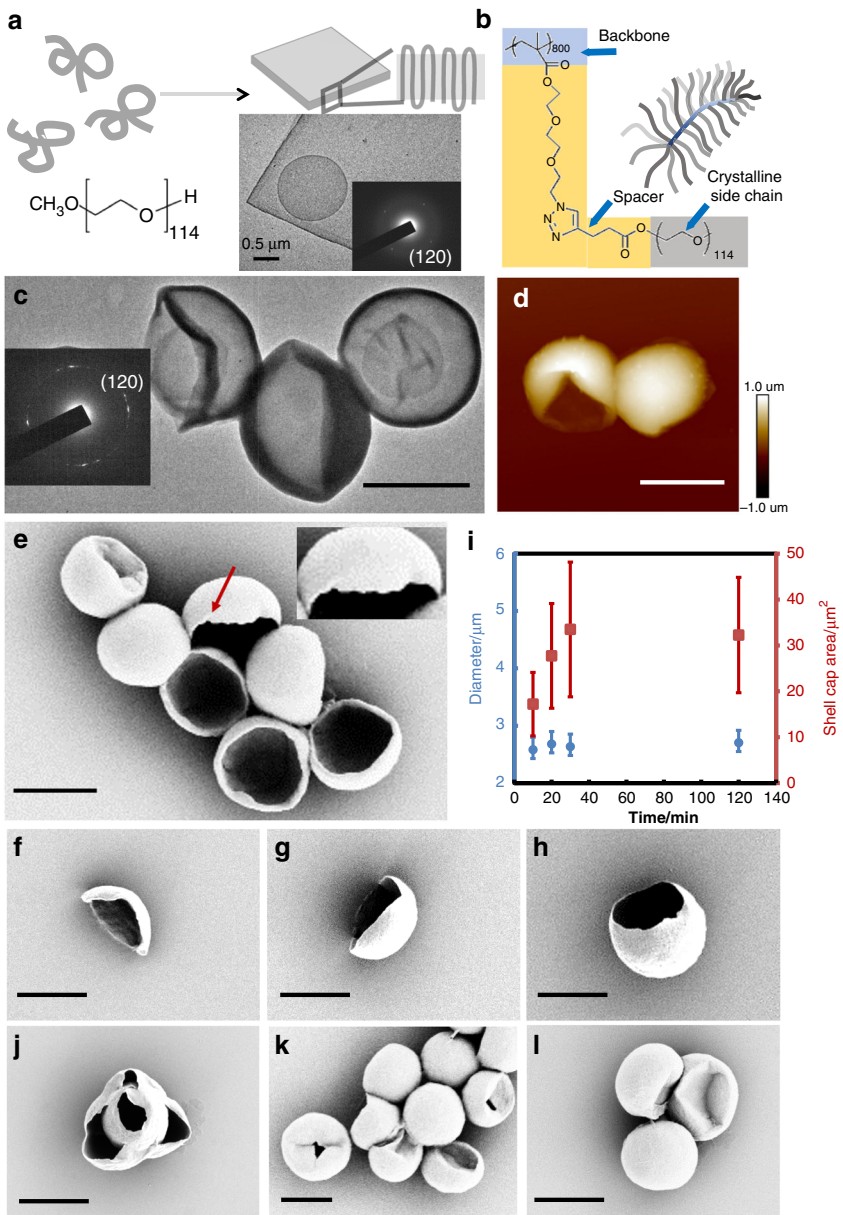

**Fig. 1 Single crystals of linear PEO and mBBs.** (**a**) Schematic illustration, TEM image and SAED pattern of a PEO PSC. (**b**) Chemical structure and schematic of an mBB. **c** TEM image with an SAED pattern, (**d**) AFM and (**e**–**h**) SEM images of mBB PSCs crystallized at 20 °C for 2 h (**c**–**e**), 10 min (**f**), 20 min (**g**) and 30 min (**h**). (**i**) Temporal evolution of mBBC. Blue dots: diameter; Red square: shell cap area; error bar represents standard deviation based on ten mBBCs. (**j**) mBB crystals from 0.02 wt.% mBB solution. (**k**, **l**) mBB crystals after adding the same mBB polymer (**k**) and 5k Da linear PEO (**l**) to pre-formed crystalline shells. Scale bars: 2 μm.

### Table 1 Molecular characteristics of mBB samples.

| Sample | DP of backbone | DP of PEO | PEO grafting density[a] | $M_{n,SEC}$ (× 10⁶ Da)[b] | PDI[b] |
|---|---|---|---|---|---|
| mBB$_{800}$-PEO$_{114}$-76 | 800 | 114 | 76% | 1.43 | 1.13 |
| mBB$_{800}$-PEO$_{114}$-75-Rh[c] | 800 | 114 | 75% | 1.46 | 1.13 |
| mBB$_{707}$-PEO$_{114}$-94 | 707 | 114 | 94% | 1.27 | 1.12 |
| mBB$_{707}$-PEO$_{114}$-48 | 707 | 114 | 48% | 1.11 | 1.16 |
| mBB$_{707}$-PEO$_{114}$-10 | 707 | 114 | 10% | 0.73 | 1.20 |

[a]PEO grafting densities were determined by using the feed molar ratio of backbone polymer and side chain polymer and the relative peak areas of the mBB and side chain polymer in the size exclusion chromatography (SEC) chromatogram of the final reaction mixture[21].
[b]The number average molar mass ($M_{n,SEC}$) and dispersity (*Đ*) for each PEO mBB sample were determined by SEC relative to linear polystyrene standards using three Mixed-B columns (Agilent Technologies) with *N,N*-dimethylformamide (DMF) containing 50 mM LiBr as eluent.
[c]A small amount of alkyne-functionalized rhodamine B (0.2 mol% with respect to backbone repeat units) was incorporated into the PEO mBB sample by simultaneously clicking with alkyne end-functionalized 5k Da PEO.

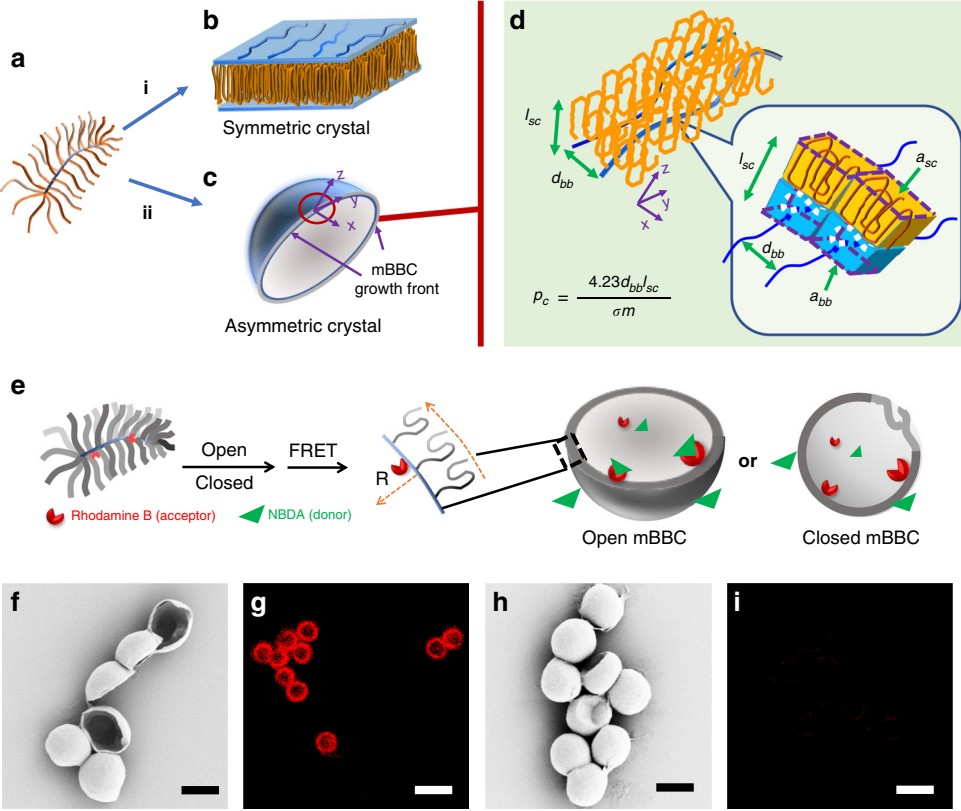

**Fig. 2 mBBC formation mechanism.** An mBB (**a**) could crystallize following pathway i to form a flat symmetric crystal (**b**). The blue shaded areas denote mBB backbones/spacers. Pathway ii leads to an asymmetric crystal (**c**, **d**), where (**d**) is an enlarged view of **c**. x and z denote growth direction and the PEO axis, respectively. Two mBB molecules are shown in (**d**). Note that each PEO side chain forms multi-stem layers (yz plane) orthogonal to the growth direction x to alleviate local packing crowdedness. The local chain crowdedness in high grafting density mBB polymers leads to cross-sectional area mismatch between the mBB backbone and side chain crystals, which induces lamella bending, translational symmetry breaking and the subsequent formation of mBBCs. The inset in (**d**) highlights one side chain per mBB molecule. The side chain crystallizes into a small yellow rectangular frustum with defined chain folding, while the blue rectangular frustum represents the volume of the closely packed amorphous layer. (**e**) Schematic illustration of FRET experimental design. (**f**, **h**) SEM images and (**g**, **i**) corrected FRET images of open (**f**, **g**) and closed (**h**, **i**) mBBCs of $mBB_{800}$-$PEO_{114}$-75-Rh. Scale bars: 2 μm in (**f**, **h**) and 5 μm in (**g**, **i**).

mBB-based spherical crystalline assemblies. As one type of crystalsomes, the mBBCs either are closed or have small openings. They differ from the classical polymersomes due to their single crystal-like nature[46]. The mBBCs also differ from linear polymer crystalsomes because in the latter case, crystals are forced to grow at a predefined spherical surface[43,44,47], while in mBBCs, no external confinement was applied, suggesting that translational symmetry is spontaneously broken during crystallization. Note that since the crystals are spherical, we can also view that the translational symmetry is recovered using a spherical coordinate to define the space.

## Formation mechanism of mBBC.
The observation of mBBCs suggests an intriguing mechanism for translational symmetry breaking: local chain overcrowding-induced lamella bending. Fig. 2 shows two possible crystallization pathways of an mBB molecule. Since the backbone is bulky and immiscible with PEO side chains, when mBB crystallizes, the backbone and the spacer would be excluded to both or one side of the lamellar crystal, leading to a symmetric lamella (Fig. 2b), or an asymmetric crystal (Fig. 2c, d) consisting of one PEO lamella (yellow) atop an mBB backbone/spacer layer (blue). In a densely grafted mBB polymer, to reduce the side chain packing density in the crystal, (1) asymmetric lamella is likely more favorable because the side chain packing is twice as crowded in the symmetric crystal compared

with the asymmetric one; (2) the local steric overcrowding can be further alleviated through bending of the crystal (Fig. 2c, d) towards the backbone layer.

To understand the detailed packing mechanism of an mBB crystal, let's start from examining the dimensions of a PEO crystal lattice. For a PEO unit cell viewing down the c axis, the projection of each unit cell along c axis (normal to the lamellar surface) is $0.86 \text{ nm}^2$, and the distance between the nearest neighboring chains is ~0.46 nm[42]. For mBB, at a grafting density of 1 PEO side chain/backbone repeat unit, the nearest neighboring tethering points are ~0.252 nm apart assuming all trans conformation of the backbone. This suggests that the backbone cannot accommodate all the side chains even if they form one layer of extended chain crystalline stems onto crystal growth front. Furthermore, PEO chains often fold as they crystallize (number of folds per side chain, ζ). In the case of mBB, folding of the PEO would significantly increase the mBB local chain crowdedness and the associated steric hindrance in the crystal (Fig. 2d). On the other hand, the backbones of mBBs, along with the spacer groups between the backbone and PEO (see the molecular structure in Fig. 1b) and a small portion of adjacent side chains, form a closely packed amorphous layer (blue rectangular frustum in Fig. 2d) with a defined inter-backbone distance to accommodate the crowded packing of side chain crystals.

We recall that the deviation from a flat interface in diblock copolymer (BCP) assembly is attributed to the asymmetric shape

of the molecule, quantified by critical packing parameter $p \equiv v/a_0 l_c$ (1), where $v$ is the volume of the hydrophobic chain, $a_0$ is the optimal area of the hydrophilic group, and $l_c$ is the critical chain length of the hydrophobic group[46,48–52]. For $p < 1/3$, $1/3 < p < 1/2$, and $1/2 < p < 1$, spherical micelles, cylindrical micelles, and vesicles are observed, respectively[53]. As a hollow vesicle, mBBC mimics the classical amphiphilic polymersome, which is an equilibrium structure in symmetric BCPs[46,48–51]. Following the original volume argument proposed by Israelachvili et al.[53], we compare the cross-sectional areas of the PEO side chain crystal layer (yellow in Fig. 2d) and the amorphous backbone layer (blue in Fig. 2d). We define $p_c \equiv a_{bb}/(v_{sc}/l_{sc}) = a_{bb}/a_{sc}$ (2), where $p_c$ is the packing parameter of the mBBC, $l_{sc}$ is the PEO crystal thickness, $v_{sc}$ is the volume of one PEO side chain in the crystal, $a_{sc}$ the cross-sectional area of one side chain in mBBC, and $a_{bb}$ is the amorphous cross-sectional area per side chain (Fig. 2d). Considering the nearly extended chain conformation of the backbone in mBBs, we can estimate $a_{bb}$ following $a_{bb} \approx d_{bb} \times l_{bb} = d_{bb} \times 0.252/\sigma$ nm$^2$ (3), where $d_{bb}$ is the inter-backbone distance, $l_{bb} = 0.252/\sigma$ is the distance between two tethering points of the adjacent side chains in the mBB with a grafting density of $\sigma$. $a_{sc}$ can be estimated as $a_{sc} = (\zeta + 1) \times 0.86/4$ nm$^2$ (4), where $\zeta$ is the fold number of PEO chain, $(0.86/4)$ nm$^2$ is projection area per chain along the $c$. Note that $\zeta$ can be calculated based on PEO DP ($m$) and the crystal thickness: $\zeta = (1.939 \times \frac{(\frac{m}{7})}{l_{sc}} - 1)$ nm (5), where 1.939 nm is the $c$ dimension, and 7 is from the $7_2$ helical conformation of the chain[14,36,54]. Taking together, we have $p_c \equiv \frac{a_{bb}}{a_{sc}} = \frac{4.23 d_{bb} l_{sc}}{\sigma m}$ (6). In BCP self-assembly, it was argued that symmetric BCPs ($p \sim 1$) leads to flat lamellae, which eventually curved into a sphere to minimize lateral free energy with decreasing $p_c$. In this case, assuming a crystalline shell with a 2μm diameter and 10 nm thickness, $p_c$ can be estimated as the ratio between inner and outer shell surface, leading to a $p_c = \frac{(R-10)^2}{R^2} \sim 0.98$ (7), where $R$ is the radius of mBBC in nm. Plugging this into Eq. (6), for $\sigma = 0.76$, $m = 114$ and $l_{sc} = 10$ nm, we can estimate $d_{bb} \sim 2.01$ $nm$, which is reasonable considering the spacer between the backbone and the PEO side chain (Fig. 1b). Note that $d_{bb}$ can be used to guide our understanding of the mBB packing, and can be significantly influenced by grafting density, side chain length and crystallization temperature. While comparing mBBCs formed by polymers with different $\sigma$, because $d_{bb}$ and $l_{sc}$ could also change, using Eqs. (6) and (7) provides a qualitive understanding of grafting density dependence of mBBCs (see later results).

The above discussion suggests that a slightly less than unity $p_c$ would lead to spherical crystals in mBBCs and that a greater $p_c$ (but still <1) corresponds to an mBBC with a larger radius (Eq. (7)). Based on this framework, several predictions can be made, (1) the mBBCs observed in Fig. 1 should be asymmetric with chemically distinctive top and bottom surfaces; (2) Eqs. (6) and (7) predict that the mBBC radius should be affected by the crystal lamellar thickness, as well as mBB $\sigma$. Increasing $l_{sc}$ or decreasing $\sigma$ would lead to a greater $p_c$ (but still <1), hence an increased mBBC radius. In the following section, we shall first experimentally confirm the asymmetry of the mBBC lamellae and then present the correlation between the mBBC size with $l_{sc}$ and $\sigma$.

**mBBC lamellae are asymmetric**. We introduced a fluorescent dye Rhodamine B on the mBB backbone (mBB$_{800}$-PEO$_{114}$-75-Rh, SI) to confirm the asymmetric nature of mBBCs. We hypothesized that upon forming mBBCs, if the overcrowding argument is correct, the crystalline PEO should be on the outer layer of the mBBC because of the crowded chain packing in the lamellae. The Rhodamine B groups would therefore be encapsulated in

the mBBC, which provides an opportunity to test the lamellar asymmetry using the FRET effect between Rhodamine B and 4-(2-acryloyloxyethylamino)-7-nitro-2,1,3-benzoxodiazole (NBDA) pair (Fig. 2e).

The FRET experiments were conducted using both open and closed mBBCs. Fig. 2f, g show the SEM and corrected confocal fluorescent microscopy images of open mBBCs and Fig. 2h, i are the closed ones. Detailed analysis can be found in the SI and Supplementary Fig. 13. The open mBBCs show a strong FRET effect (Fig. 2g), while the closed ones show none (Fig. 2i). For open mBBCs, free NBDA can diffuse to the vicinity of the Rhodamine B groups, and FRET could occur. While for closed mBBCs, the crystalline shell is a strong diffusion barrier for NBDA[44], preventing close contact between the FRET donor and acceptor, minimizing the FRET effect. These experiments confirmed that the Rhodamine B groups on the mBB backbone were excluded to the inner surface of mBBCs while the PEO crystals occupied the outer layer, leading to the unbalance of inner and outer surfaces of the crystals.

The asymmetric nature of the mBBC shells is the origin of lamellar bending. Similar observations have been reported in PVDF and polyamide 66 scrolled single crystals[8,13], where the unbalanced folding in these two cases leads to lamellar asymmetry and the subsequent crystal bending. In triblock copolymer polystyrene-*b*-poly (ethylene oxide)-*b*-poly (1-butene oxide) (PS-*b*-PEO-*b*-PBO), upon PEO crystallization, PS and PBO, respectively, separate into top and bottom surfaces of the PEO single crystal, leading to crystal bending. Note that in all these cases, lamellae are considered as a three-layer structure with a crystalline central layer while the top and bottom layers are either fold surface (PVDF and polyamide 66) or PS/PBO domains. In this study, we use a two-layer model, which introduces similar asymmetry that can lead to lamellar bending. The asymmetric nature of the mBBC shell can also be supported by a thermodynamic argument, detailed in the supporting information (Supplementary Fig. 14).

**mBBC size is lamellar thickness-dependent**. mBBC shell thickness can be varied by changing $T_c$ (20, 25, and 30 °C). As shown in Fig. 3, mBBCs were grown at all three $T_c$ with a diameter of 2.70, 3.14, and 3.44 μm, respectively (Fig. 3g). To measure the crystal thickness, mBBC samples were collected at early stages of the growth before significant bending occurred for AFM imaging (Fig. 3d–f). The measured thicknesses are 9.7, 10.5, and 12.7 nm, indicating thicker lamellae in larger mBBCs. This positive correlation between lamellar thickness and mBBC diameter is consistent with the packing parameter prediction and our chain overcrowding argument; as the lamellar thickness increases, the chain packing within the crystals becomes less crowded and the packing stress can therefore be alleviated, leading to a larger mBBC.

**mBBC size is grafting density-dependent**. Direct control of the local chain crowdedness can also be achieved by tuning $\sigma$. A series of mBBs with three different $\sigma$ was synthesized, namely, mBB$_{707}$-PEO$_{114}$-94, mBB$_{707}$-PEO$_{114}$-48, and mBB$_{707}$-PEO$_{114}$-10. PSCs were grown using $T_c$ of 20, 25, and 30 °C. As shown in Fig. 4a–f, mBBCs were successfully formed in mBB$_{707}$-PEO$_{114}$-94 and mBB$_{707}$-PEO$_{114}$-48. Fig. 4g shows that at a constant $T_c$, when $\sigma$ decreased from 0.94 to 0.48, the mBBC diameter increased for all $T_c$. When $\sigma$ further decreased to 0.1 for mBB$_{707}$-PEO$_{114}$-10, mBBCs were not observed; the crystals were flat with a 2D morphology (Fig. 4h). All these results are consistent with our side chain overcrowding framework. For $\sigma = 0.94$, nearly every other carbon atom is tethered with a side chain. For $\sigma = 0.48$, ~4

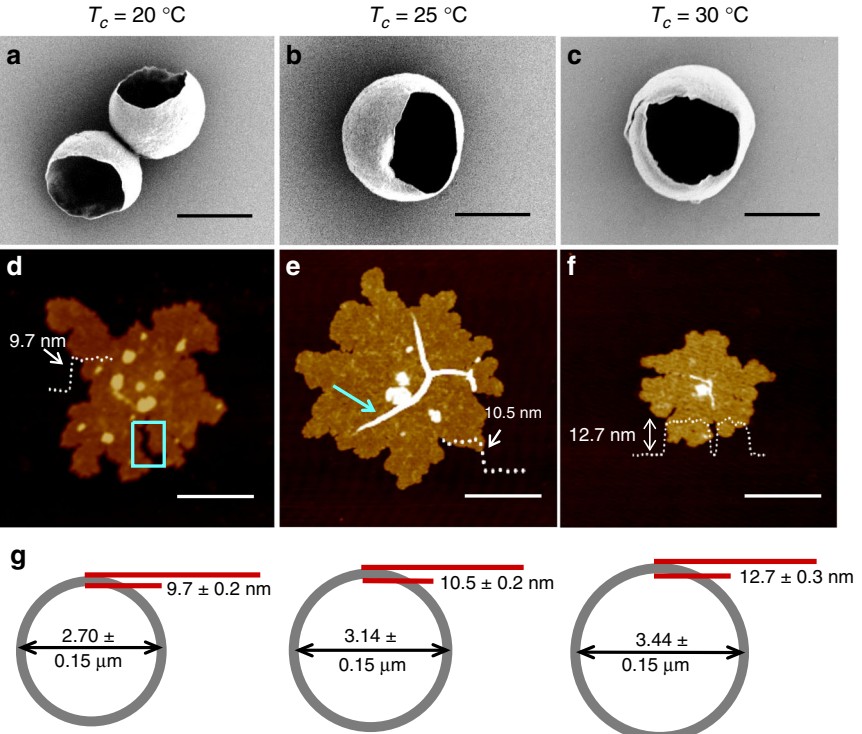

**Fig. 3 Manipulating PEO mBBC sizes via $T_c$.** (a–c) SEM images of mBBCs and (d–f) AFM images of small mBBC pieces crystallized at $T_c = 20$ (a, d), 25 (b, e) and 30 °C (c, f). (g) Schematics depicting the correlation between mBBC size and shell thickness; value = average ± standard deviation (based on measurements of ten mBBCs). The box in (d) and arrow in (e) respectively show a crack and a corrugation formed due to flatening of the crystals. Scale bars: 2 μm in (a–c), 0.5 μm in (d–f).

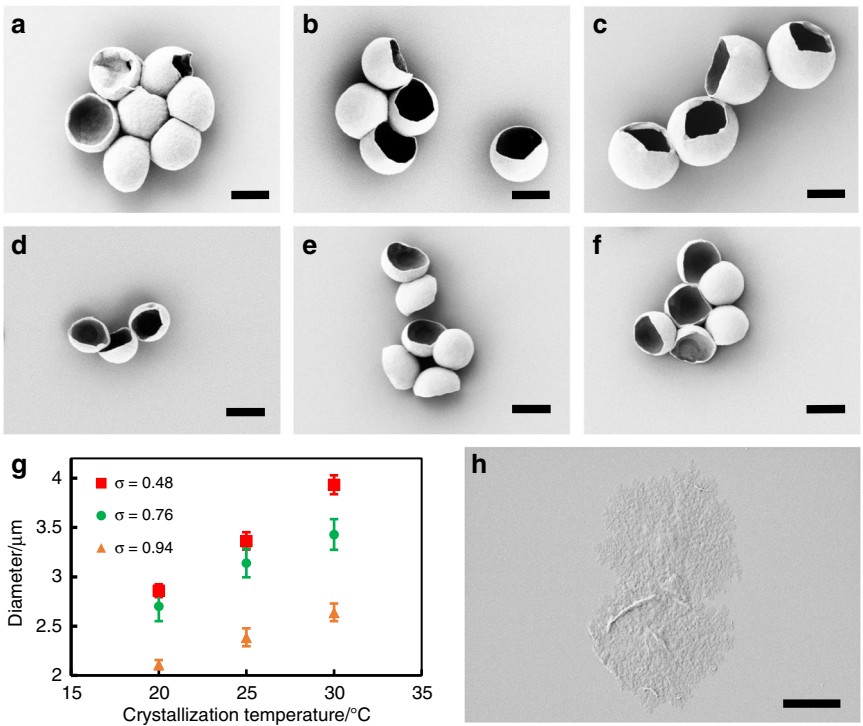

**Fig. 4 Manipulating PEO mBBC sizes via grafting density.** (a–f) SEM images of mBBCs of mBB$_{707}$-PEO$_{114}$-48 (a–c) and mBB$_{707}$-PEO$_{114}$-94 (d–f), $T_c = 20$ (a, d), 25 (b, e) and 30 °C (c, f). (g) Plot of mBBC diameter vs. $T_c$; error bar represents standard deviation based on measurements of ten mBBCs. (h) SEM image of mBB$_{707}$-PEO$_{114}$-10 crystal. Scale bars: 2 μm.

backbone carbon atoms bear one side chain. Due to the long spacer group between PEO and the backbone (16 atoms as shown in the mBB chemical structure in Fig. 1b) and side chain folding, the distribution of the PEO side chains along the backbone, we believe, does not significantly affect the mBBC crystallization in the medium to high grafting density ($\sigma \geq 0.48$), and mBBCs were observed.

## Discussion

We observed spontaneous translational symmetry breaking in the crystallization of PEO mBBs. In contrast to 2D flat lamellar crystals of linear PEO, the mBBs with $\sigma \geq 0.48$ crystalized into spherical mBBCs with broken translational symmetry. FRET experiments demonstrated that the mBBC shell is asymmetric with crystalline PEO side chains folded in the outer layer of the mBBCs. This asymmetric packing arises from local side chain overcrowding of the mBB molecules, evidenced further by the absence of mBBCs in low $\sigma$ mBBs. A packing parameter quantifying the crystalline lattice and the backbone cross-sectional area was introduced to rationalize the curvature observed in mBBCs. The mBBCs were formed spontaneously upon crystallization, allowing for a versatile structural design towards functional nanomaterials.

## Methods

**Materials.** Pentyl acetate was purchased from Sigma-Aldrich and distilled before use. NBDA was synthesized as reported in a previous publication[21]. Poly(ethylene glycol) monomethyl ether (number average molar mass, $M_n = 5k$ Da) was purchased from Sigma-Aldrich. PEO molecular bottlebrushes were synthesized using a "grafting to" method (see Supplementary Fig. 1 and the section of synthesis of PEO molecular bottlebrushes). Table 1 in the main text lists all PEO mBB samples that were studied in this report.

The synthesis of azide-functionalized backbone polymer PTEGN$_3$MA with a DP of 800 (PTEGN$_3$MA-800) was reported in a previous publication[21]. Another backbone polymer PTEGN$_3$MA with a DP of 707 (PTEGN$_3$MA-707) was prepared by using the same procedure as for PTEGN$_3$MA-800. $^1$H NMR spectroscopy analysis indicated that the degree of azide-functionalization for PTEGN$_3$MA-707 was 99%. SEC analysis results for PTEGN$_3$MA-707: $M_{n,SEC} = 307,100$ Da; $Đ = 1.10$ (relative to linear polystyrene standards). The alkyne end-functionalized PEO was synthesized by an $N$-(3-dimethylaminopropyl)-$N'$-ethylcarbodiimide hydrochloride-catalyzed coupling reaction between poly(ethylene glycol) monomethyl ether with a molar mass of 5k Da (CH$_3$O-PEO-OH) and 4-pentynoic acid, as described previously[21]. $N,N,N',N'',N''$-Pentamethyldiethylenetriamine (PMDETA, 99%, Acros) was purified by vacuum distillation over calcium hydride. All other chemicals were purchased from either Aldrich or Fisher and used as received.

**Methods.** SEC of PEO molecular bottlebrush samples and PTEGN$_3$MA-707 was carried out at 50 °C using a PL-GPC 50 Plus (an integrated GPC/SEC system from Polymer Laboratories, Inc.) with a differential refractive index detector, one PLgel 10 μm guard column (50 × 7.5 mm, Agilent Technologies), and three PLgel 10 μm Mixed-B columns (each 300 × 7.5 mm, linear range of molecular weight from 500 to 10,000,000 Da according to Agilent Technologies). DMF with 50 mM LiBr was used as the mobile phase at a flow rate of 1.0 mL min$^{-1}$. The SEC system was calibrated with a set of narrow disperse linear polystyrene standards (Scientific Polymer Products, Inc.), and the data were processed using Cirrus$^{TM}$ GPC/SEC software (Polymer Laboratories, Inc.). For grafting density analysis, the same SEC system, except with the use of a PSS GRAL guard column (50 × 8 mm) and two PSS GRAL columns (each 300 × 8 mm, linear molecular weight range from 500 to 1,000,000 Da) instead of Mixed-B columns, was employed. $^1$H and $^{13}$C NMR spectra were recorded on a Varian Mercury 300 or a Varian VNMRS 500 NMR spectrometer, and the residual solvent proton signal was used as the internal standard.

In microscopy sample preparation, a drop of 10 μL mBB crystalsome solution was drop cast onto piranha-cleaned cover slides (AFM/SEM) or carbon-coated copper grid (TEM), and then dried overnight under vacuum. Before imaging, the sample was coated with Pt/Pd (SEM) or carbon (TEM). SEM images were taken on a ZEISS Supra 50VP microscope with a 1 kV accelerating voltage. TEM images were obtained under a JEOL 2100 microscope with a 120 KV accelerating voltage. AFM images were acquired using a Bruker multimode 8 AFM with a tapping mode. Fluorescence imaging was performed by using an Olympus FV1000 confocal system. FRET package embedded OLYMPUS FLUOVIEW was used to obtain, process images and generate corrected FRET images. Raw images were converted to tiff files with pseudocolors for display.

**Synthesis of PEO molecular bottlebrushes.** Five PEO molecular bottlebrush samples were prepared by grafting alkyne end-functionalized PEO with a molar mass of 5k Da onto the azide-functionalized backbone polymer, either PTEGN$_3$MA-800 or -707, via copper(I)-catalyzed azide-alkyne cycloaddition reaction: two from PTEGN$_3$MA-800 and three from PTEGN$_3$MA-707 (see Table 1 in the main text). Detailed below is the synthetic procedure for mBB$_{800}$-PEO$_{114}$-76. All other bottlebrush samples were prepared using the same procedure. For mBB$_{800}$-PEO$_{114}$-75-Rh, alkyne-functionalized Rhodamine B (RhB-alkyne), which was synthesized from Rhodamine B and propargyl alcohol, was incorporated into the molecular brushes using a feed molar ratio of 0.2% of RhB-alkyne with respect to backbone repeat units.

Backbone polymer PTEGN$_3$MA-800 (5.06 mg, 0.0208 mmol repeat units, from a stock solution in tetrahydrofuran, THF) was weighed out into a 3.7 mL vial equipped with a stir bar. THF was evaporated off with a gentle stream of nitrogen, and the polymer was redissolved in DMF (0.5 mL). Alkyne end-functionalized PEO with a molar mass of 5k Da (211.6 mg, 0.0415 mmol) was weighed into a separate vial, dissolved in DMF (1.0 mL), and transferred to the vial containing PTEGN$_3$MA-800 using additional DMF (1.0 mL) to rinse. CuCl (2.5 mg, 0.025 mmol) was added. A rubber septum was attached, and the vial was flushed via needles with nitrogen before PMDETA (5 μL, 0.024 mmol) was injected via microsyringe. The reaction progress was monitored by SEC. After 20 h, propargyl alcohol (50 μL, 0.86 mmol) was injected to cap any unreacted azide units. (For the bottlebrushes prepared using PTEGN$_3$MA-707, benzyl propargyl ether was used instead of propargyl alcohol.) The mixture was stirred for an additional 2 h before the reaction was stopped by passing through a short neutral alumina/silica gel column with CH$_2$Cl$_2$ as eluent to remove the catalyst. The unreacted side chains were removed by centrifugal filtration (50k Da MWCO) in water. The purified brushes were dried under high vacuum (yield: 68 mg) and dissolved in THF for storage. The grafting density was determined to be 76% by comparison of the brush and the unreacted side chain polymer peak areas from the SEC chromatogram of the mixture at the end of the reaction. The complete removal of PEO side chains was confirmed by SEC analysis using a PL-GPC 50 Plus system (PSS GRAL columns, linear range of molecular weight from 500 to 1,000,000 Da) with DMF containing 50 mM LiBr as the mobile phase (Supplementary Fig. 2A). SEC analysis results for the purified brushes mBB$_{800}$-PEO$_{114}$-76 using a PL-GPC 50 Plus system (Agilent Mixed-B columns) with DMF/50 mM LiBr as the mobile phase: $M_{n,SEC} = 1.43 \times 10^6$ Da; $Đ = 1.13$ (Supplementary Fig. 2B).

**Crystallization of mBB crystalsome.** As-synthesized PEO mBBs were dissolved in dry THF and stored in a freezer to prevent degradation. Uniform PEO mBB crystalsomes were obtained using a self-seeding method, and the temperature profile is shown in Supplementary Fig. 7[54,55]. In brief, a 0.01 wt.% PEO mBB solution was first prepared by dissolving vacuum-dried PEO mBB (from its THF solution) in distilled pentyl acetate at 85 °C for an hour. The solution was stored at −10 °C for at least 12 h and then brought to a seeding temperature for 12 min to obtain crystal seeds. Detailed seeding temperatures for different PEO mBB samples can be found in the main text. The seed-containing solution was then allowed to crystalize at different crystallization temperatures (20/25/30 °C) to develop PEO mBBCs. After crystallization, any remaining uncrystallized PEO mBB was removed by centrifugation and the resultant mBBCs were redispersed in pentyl acetate at a concentration of 0.01 wt.% before further use.

**Nucleation control through controlling seeding temperatures.** To generate different seed contents, the solution of mBB$_{800}$-PEO$_{114}$-76 was seeded at 44.1, 44.3, 44.5, 44.8, and 45.0 °C for 12 min before crystallized at 20 °C. A control experiment of crystallization without employing self-seeding was conducted by directly quenching a fully dissolved mBB$_{800}$-PEO$_{114}$-76 solution to 20 °C. Supplementary Fig. 8 shows the self-seeding effect on crystal growth.

**AFM study of mBBC.** A small mBB$_{800}$-PEO$_{114}$-76 crystal was obtained by using a 44.1 °C seeding temperature and a crystallization time of 15 min. Water bath sonication was applied to the crystal solution for 5 s. The sample solution was drop cast onto a clean glass cover slip and dried with a stream of nitrogen before AFM imaging (Supplementary Fig. 12).

**Temporal evolution of mBBC formation.** After seeding the mBB$_{800}$-PEO$_{114}$-76 solutions at 44.8 °C for 12 min, the solutions were brought to 20 °C for 10, 20 and 30 min. Intermediate structures were collected by immediately centrifuging the above-mentioned solutions at 3500 rpm for 5 min and re-dispersing the solids in pentyl acetate.

**Programmed growth to achieve closed PEO mBBCs.** PEO 5k Da homopolymer was fully dissolved in 60 °C pentyl acetate at 0.01 wt.% for 10 min and the solution was allowed to cool to room temperature. A 50 μl of the PEO solution was added into 200 μL mBB$_{800}$-PEO$_{114}$-76 or mBB$_{800}$-PEO$_{114}$-75-Rh crystalsome suspensions. The mixture was then placed at room temperature for another hour for crystallization.

**FRET study**. NBDA, a FRET donor for Rhodamine B was dissolved in pentyl acetate with a concentration of 0.007 mg g$^{-1}$. mBBCs of mBB$_{800}$-PEO$_{114}$-76 and mBB$_{800}$-PEO$_{114}$-75-Rh were dispersed in pentyl acetate with a concentration of 0.1 mg g$^{-1}$. FRET experiments on opened mBB$_{800}$-PEO$_{114}$-75-Rh and closed mBB$_{800}$-PEO$_{114}$-75-Rh crystalsome samples were correspondingly conducted using an Olympus FV1000 fluorescence microscope. In the experiment, the samples were prepared by mixing the mBB$_{800}$-PEO$_{114}$-75-Rh crystalsome solution and the NBDA solution at a volume ratio of 1:1. Donor-only sample was prepared by mixing the mBB$_{800}$-PEO$_{114}$-76 crystalsome solution and the NBDA solution at a volume ratio of 1:1. Acceptor-only sample was the mBB$_{800}$-PEO$_{114}$-76-Rh crystalsomes solution mixed with 1:1 pentyl acetate. Each of the above mixtures was drop cast onto cover slide and dried before taking fluorescence images under different setup. Supplementary Fig. 12 shows the FRET results.

**Reporting summary**. Further information on research design is available in the Nature Research Reporting Summary linked to this article.

## Data availability

All data are available in the main text and the supplementary materials.

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

## Acknowledgements

This work was supported by the National Science Foundation Grant DMR 1709136 and DMR 1607076. Partial funding for open access to this research was provided by University of Tennessee's Open Publishing Support Fund.

## Author contributions

C.Y.L. and B.Z. designed the research; H.Q. and X.T.L. fabricated and characterized crystalsomes; D.M.H. synthesized the polymers. S.M. and M.C.S. characterized the crystalsomes. C.Y.L., H.Q., X.T.L., D.M.H., and B.Z. analyzed the data and wrote the manuscript.

## Competing interests

The authors declare no competing interests.
