## [Peer Review File · Nature Communications]

Reviewers' comments:

Reviewer #1 (Remarks to the Author):

Qi et al. present a study on the crystallization of a PEO-based bottlebrush polymer, where a spacer separates the PEO chains from the backbone chain and the level of substitution of the PEO onto the main chain is varied. Using a self-seeding technique at well-defined temperatures and for specific (dilute) concentrations of the bottlebrush polymer the crystal growth habit is spherical due, according to the authors, to a crowding of the side chains by the main chain that forces the PEO crystals (all non-folded) into a spherical habit. This communication raises more questions, in my opinion, than answers, and I feel that the work is premature for publication and more experiments are necessary before the work is suitable.

From the electron diffraction data in the inset of Figure 1c the diffraction profiles are circular which indicate that the electron beam is intercepting multiple smaller crystals within the spherical shell. These data would suggest that the crystal lattice is not distorted, since that would give a spreading of the diffraction profile in q , not in the azimuthal direction. So, Am I to take from these data that the shells are comprised of multiple smaller crystals patched together in the sphere? Is there evidence for broadening in q ? If so, this data should be shown.

The authors make a big deal out of breaking translational symmetry. This is true if you stay with a Cartesian coordinate system. What if I changed to a spherical coordinate system, since that would be a fairer treatment. The use of the phraseology "Defying" is simply hype that should be removed.

If I am not mistaken, there is no chain folding in the crystals, since this would exacerbate the crowding at the junction with the spacer/main chain. How do the melting points of these PEO crystals compare with PEO crystallized in the bulk, with constraints? If this junction point is so crowded, I would have expected that the dilution of the number of PEO chains by a factor of 2 or 3 would be sufficient to alleviate this crowding. However, planar crystals are not observed until the dilution is a factor of 10. I was very surprised by this result and would like to see a more quantitative analysis and, also, an analysis of the randomness of the substitution along the chain. I am certain this must be playing a role but there was no discussion of this. A quantitative analysis of the "randomness" of substitution along the chain is necessary since that is very important for understanding the data and the arguments of the authors

"mysterious snowflakes and dazzling quartz" please cut the hypes out. There is no need for this and it is distracting.

If I am not mistaken from the results shown, it is NOT possible to get a completely spherical shell of the crystalline bottlebrush copolymers. The authors report that the crystallization proceeds and the supply of the bottlebrush is exhausted. Increasing the initial concentration results in the rose petal habit. The addition of extra bottlebrush polymer does not allow completion of the sphere. This result is rather baffling and I simply do not believe the argument that it is too difficult for the bottlebrush copolymer to enter into the shell to complete crystallization. There have been countless works of flexible polymer diffusing through nanoscopic pores and the hole remaining in the crystals presented are simply much too large. There must be another reason. To complete the spheres, the authors cheat and use a linear PEO and, while it seals the hole, it is far from a sphere and nothing is said about the PEO crystals when they eventually meet. Certainly, these contacts must be rife with distortions or maybe even holes. The absence of any statements is glaring.

The self-seeding step is absolutely critical for the authors to observe the spherical habit. Yet, we, the readers are left with this being a magical step in the process and I consider this to be one of the most important parts of this manuscript that is left completely open. What is the nature of the seed crystal? It is at this stage that the die is cast. In order to comprehend the manuscript fully, a deeper discussion of the seed crystals is necessary.

The authors make arguments about the crowding giving rise to the observed spherical habit. Yet, they also make a statement about the immiscibility of the backbone with spacer and the PEO side chains. Wouldn't this play a critical role in segregating the PEO chains to enable the crystallization to proceed in the manner that it does? Some thermodynamic arguments about the state of

miscibility at the formation of the seed crystals is in order. One need to take the PEO of both sides of the brush, bring them to one side (which will require energy) and then get them crystalize. It is at this stage where the growth begins that templates everything else.

“ where micro-faceted edges (red arrows in Figures 1E) further confirms the crystalline nature “
Doesn't this say there must be a registry of the crystals and, if this were in 2D, there would be no translational symmetry issues. I might add that, given the density of data put into each figure, the micrographs are bordering on the verge of being too small to discern the edges of the spherical shells where this faceting is evident. A magnification of this would have helped.

In Figure 1 there is a schematic of the brush polymer that one sees everywhere. That is fine but I would like to have seen a similar schematic for a single chain in the crystals that are formed. Certainly, the side chains must be on one side of the main chain. Along these lines, I have gotten no sense of the behavior of the main chain when the system crystallizes. For the planar single crystals somewhat, linear squiggles are shown in Fig.2C. Do the authors mean to infer that the backbones are stretched out in that manner. For the spherical crystals, even that is not provided. However, the configuration of the backbone chain must be an important consideration, yet it is left untouched. This must also be an important consideration when the authors tried to complete the spheres with the addition of more bottlebrush polymer (that failed). I am really left wanting without any information on the backbone chains and that is certainly critical for the crowding arguments as well.

If I were to consider all the PEO chains in the bottlebrush polymers, what fraction of the chains are incorporated into the crystals?

“The measured thicknesses are 9.7, 10.5 and 12.7 nm, indicating thicker lamellae in larger mBBCs . This also argues that there is no chain folding, since there would be an integral number of stems. However, I ask, what happens to the melting points and does that change systematically with crystal thickness?

Reviewer #2 (Remarks to the Author):

The manuscript by Christopher Li and co-authors describes a very interesting case of non-planar crystals formed by the bottlebrush polymers with crystallizable side chains. The studied crystals have a very unusual shape of hollow spheres. The authors emphasize the absence of translational symmetry for such objects claiming that this case exemplifies a new principle of spontaneous translational symmetry breaking.

The paper is very interesting and certainly deserves publication. However, several issues have to be addressed before it is accepted:

1. The authors would like to provide more details about previous reports on non-planar polymer crystals. In particular, I would propose to be more specific in describing the known crystal habits such as helicoids, scrolls, 3D helices as well as the experimental methods allowing to address the details of these morphologies including synchrotron-based micro- and nano-focus X-ray scattering.
2. In relation to p.1, it would be desirable if the authors can propose a structural model explaining the morphogenesis of such spherical-shaped crystals. It will be worth comparing the formation of all main non-planar crystal habits together with the newly discovered hollow spheres.
3. The referee would like to learn more about the mechanics of such objects. Can the authors rationalize the fact that they preserve their 3D shape after deposition on the substrate. What are the conditions for them to keep their 3D shape.

Reviewer #3 (Remarks to the Author):

This communication reports an unusual crystallization behavior of bottlebrush polymer with crystallizable side chains. Crystallization-induced self-assembly of PEO brushes creates microspheres with a curvature controlled by grafting density. Lower grafting density would lead to larger 'crystalsomes'. This finding is significant and merits publication.

The central hypothesis of this paper is segregation of all side chains on one side of crystalline lamellae. This segregation results in spontaneous curvature and ultimately yields spherical shells. This hypothesis makes sense, however, needs better justification. The crowdedness argument is relevant to conformation of brush molecules in solution. Crystallization morphology is largely controlled by kinetics and short-range interactions. The backbone plays role of a kinetic constraint and a packing defect. Such defects tend to segregate at the surface of crystallites. In sum, the origin and molecular mechanism of curvature is unclear to me.

Minor comments:

1. The introduction signifies the novelty of this work by comparing it to chiral and bending polymer crystalline domains. However, the two paragraphs can be better streamlined if the author would focus on specific driving forces behind crystallite curvature.
2. Fig 2d is misleading as it suggests each side chain forms its own domain
3. The purpose of introducing Eq.6 needs justification. From its current state, dbb can only be calculated from pc and not from measurement, making this equation unverifiable with data presented in this work. Meanwhile, any changes to the chain architecture or thermal history will also change dbb , making the predictions to self-assembly behavior inaccurate.
4. The outcome of tests regarding lsc and σ proves the theory that the backbone and sidechain phase separates and the side chains are on the outside and less dense side chain would result in less curvature. However, the changes are not on a similar scale, contrary to the suggestions of prediction 2 from eq.6. As an example, when comparing samples with $\sigma=0.48$ vs 0.94 . σ changed by $\sim 100\%$ yet pc changed by 0.4% . Obviously, dbb and/or lsc also changed. Though the trend predicted is correct, it cannot be deducted quantitatively from eq.6.

Reviewers' comments:

Reviewer #1 (Remarks to the Author):

Qi et al. present a study on the crystallization of a PEO-based bottlebrush polymer, where a spacer separates the PEO chains from the backbone chain and the level of substitution of the PEO onto the main chain is varied. Using a self-seeding technique at well-defined temperatures and for specific (dilute) concentrations of the bottlebrush polymer the crystal growth habit is spherical due, according to the authors, to a crowding of the side chains by the main chain that forces the PEO crystals (all non-folded) into a spherical habit. This communication raises more questions, in my opinion, than answers, and I feel that the work is premature for publication and more experiments are necessary before the work is suitable.

R: We appreciate the reviewer's comments and have conducted suggested experiments to address the reviewer's questions.

1. From the electron diffraction data in the inset of Figure 1c the diffraction profiles are circular which indicate that the electron beam is intercepting multiple smaller crystals within the spherical shell. These data would suggest that the crystal lattice is not distorted, since that would give a spreading of the diffraction profile in q , not in the azimuthal direction. So, Am I to take from these data that the shells are comprised of multiple smaller crystals patched together in the sphere? Is there evidence for broadening in q ? If so, this data should be shown.

R: To convert a crystalline lattice from a flat surface to a spherical shell, a crystalline lattice has to splay along the spherical surface, as reported in our previous work (*Nat. Commun.* **2016**, *7*, 10599; *Nanoscale* **2018**, *10*, 268.). The following schematic representation shows the a , b directions of a lattice splay at the bottom of a sphere. The rectangles in the figure represent a 2D unit cell in different locations of the shell. Unit cell a , b axes gradually change directions as we move away from the center.

This type of continuous lattice splaying can be illustrated in the entire surface for a spherical crystal, as shown in the following schematic representation:

Because of the lattice splaying, multiple domains could satisfy diffraction conditions in a selected area electron diffraction experiment (*e.g.* two circled domains in the above schematic), and diffractions from these domains would have different lattice orientations (*e.g.* the *a* axes in the circled domains have different directions), leading to arcing of diffraction spots along the azimuthal direction. The diffraction arcs appear similar to the case of a multiple-domain structure.

Diffraction peak broadening along *q* direction has been reported in crystalsome systems (*Nat. Commun.* **2016**, 7, 10599). The broadening is attributed to the limited space for the crystal to grow before bending along the spherical surface. We measured wide angle X-ray diffraction pattern of flat PEO single crystals and the mBBCs (we did not use the electron diffraction results because the potential electron beam damage in the experiments would affect detailed analysis). Due to lattice strain from curvature, the X-ray diffraction peaks of PEO mBBC showed peak broadening compared with a flat PEO single crystal, as shown in the following figure.

The average crystallite size can be calculated from full widths at half maximum (FWHM) of each diffraction peak using Scherrer equation. Compared with flat PEO single crystals, the average crystallite size of the PEO mBBCs decreases, indicating that smaller crystallites were formed to accommodate the curved space.

PEO Crystals	FWHM (°)/Crystal size (nm) orthogonal to (120), $2\theta=19.21^\circ$	FWHM (°)/Crystal size (nm) orthogonal to (032), $2\theta=23.40^\circ$
PEO BBC	0.62/12.7	0.89/8.89
PEO single crystal	0.37/21.1	0.73/10.74

2. The authors make a big deal out of breaking translational symmetry. This is true if you stay with a Cartesian coordinate system. What if I changed to a spherical coordinate system, since that would be a fairer treatment. The use of the phraseology “Defying” is simply hype that should be removed.

R: We agree that if a spherical coordinate system is used, we can define a new “spherical translational symmetry” to describe the observed mBBC crystal. In classical crystallography, the translational symmetry is defined based on a Cartesian coordinate system. We chose to use this classical definition because it allows us to better discuss lattice packing and compare the present case with conventional crystals. We changed the title to “**Breaking** translational symmetry via polymer chain overcrowding in molecular bottlebrush crystallization” in the revised manuscript.

3. If I am not mistaken, there is no chain folding in the crystals, since this would exacerbate the crowding at the junction with the spacer/main chain. How do the melting points of these PEO crystals compare with PEO crystallized in the bulk, with constraints? If this junction point is so crowded, I would have expected that the dilution of the number of PEO chains by a factor of 2 or 3 would be sufficient to alleviate this crowding. However, planar crystals are not observed until the dilution is a factor of 10. I was very surprised by this result and would like to see a more quantitative analysis and, also, an analysis of the randomness of the substitution along the chain. I am certain this must be playing a role but there was no discussion of this. A quantitative analysis of the “randomness” of substitution along the chain is necessary since that is very important for understanding the data and the arguments of the authors

R: PEO side chains do fold in the mBBC crystals, and the fold number can be calculated using equation 5 in the manuscript: $\zeta = (1.939 \times \frac{m}{l_{sc}} - 1)$, where ζ is the fold number, m is the degree of polymerization of PEO side chain, l_{sc} is the lamellar thickness. The molecular weight of the PEO side chain is 5000 g/mol, PEO degree of polymerization is $5000/44 \approx 114$, and the measure l_{sc} ranges from 9.7-12.7 nm. ζ can therefore be calculated to be ~ 2.3 to 1.5, indicating that each side chain folds 1.5 to 2.3 times. This also indicates that non-integral folding occurred in the system, which is not uncommon for 5000 g/mol PEO.

To obtain the melting points of mBBCs, we conducted differential scanning calorimetry experiments on mBBCs with the crystallization temperature (T_c) of 20, 25, 30 degree Celsius. The first heating, first cooling and second heating results are shown in the following figure and table. The melting point from second heating curve can be considered as the melting point of bulk molecular bottle brushes. From the table, we can conclude that when increasing the crystallization temperature, mBBC melting points are similar for $T_c = 20$ and 25 °C (0.1 °C difference), slightly lower than that of $T_c = 30$ °C (0.3-0.4 °C). The slight melting point difference may be because of the differences of crystal lamellar thickness and the mBBC diameter at different T_c . mBBC melting points are slightly higher than the melting point of bulk mBBCs (~ 0.4 -0.9 °C). This can be explained that the bulk melting point was measured for mBB crystals formed during the first cooling run, and the crystals may be less stable than those formed in the solution crystallization. Bulk crystallization and melting temperatures can be influenced by cooling rate and crystallization kinetics. We will study this interesting question in detail in our future work.

DSC scan of mBBCs crystallized at different temperatures. (a) first heating, (b) first cooling, (c) second heating.

Crystallization temperature	First heating melting temperature	Second heating melting temperature
20 °C	56.2 °C	55.8 °C
25 °C	56.1 °C	55.5 °C
30 °C	56.5 °C	55.6 °C

The side chains were grafted to the backbone *via* a grafting-to method. Considering the extremely large side groups (the chemical structure of the mBB molecule is redrawn on the left), it is anticipated the side chains are randomly grafted onto the backbone because if we assume

there is a segment of backbone having significantly lower grafting density, there then would have another segment of backbone bearing significantly higher grafting density, which is unlikely due to steric hindrance imposed by the side chains. mBBCs were observed with grafting density of 0.4 to 0.94. For 0.9, nearly every other carbon atom is tethered with a side chain. For a grafting density of 0.48, approximately 4 backbone carbon atoms bear one side chain. Due to the long spacer group between PEO and the backbone (16 atoms as shown on the left) and side chain folding, the distribution of the side group, we believe, does not significantly affect the mBBC crystallization.

We added: “For $\sigma = 0.94$, nearly every other carbon atom is tethered with a side chain. For $\sigma = 0.48$, approximately 4 backbone carbon atoms bear one side chain. Due to the long spacer group between PEO and the backbone (16 atoms as shown in the mBBC chemical structure) and side chain folding, the distribution of the side group, we believe, does not significantly affect the mBBC crystallization in the medium to high grafting density region ($\sigma \geq 0.48$), and mBBCs were observed.” On page 19, line 7 from the bottom.

4. “mysterious snowflakes and dazzling quartz” please cut the hypes out. There is no need for this and it is distracting.

R: We changed “mysterious snowflakes to dazzling quartz” to “snowflakes to quartz”.

5. If I am not mistaken from the results shown, it is NOT possible to get a completely spherical shell of the crystalline bottlebrush copolymers. The authors report that the crystallization proceeds and the supply of the bottlebrush is exhausted. Increasing the initial concentration results in the rose petal habit. The addition of extra bottlebrush polymer does not allow completion of the sphere. This result is rather baffling and I simply do not believe the argument that it is too difficult for the bottlebrush copolymer to enter into the shell to compete crystallization. There have been countless works of flexible polymer diffusing through nanoscopic pores and the hole remaining in the crystals presented are simply much too large. There must be another reason. To complete the spheres, the authors cheat and use a linear PEO and, while it seals the hole, it is far from a sphere and nothing is said about the PEO crystals when they eventually meet. Certainly, these contacts must be rife with distortions or maybe even holes. The absence of any statements is glaring.

R: As we described in the manuscript, a few strategies had been explored to prepare a closed shell. By increasing molecular bottle brush concentration or adding molecular bottle brushes stepwise, under the present experimental conditions (solution crystallization at three crystallization temperatures), completely closed mBBCs were not observed. As the reviewer pointed out, there have been many works on translocation of a polymer chain through nanopores.

In the present case, molecular bottle brushes have a large size and the chains have to align themselves in specific orientation for crystallization, which could be difficult in a limited space such as nanosized pore. It is reasonable that crystallization slows down as the opening of the mBBCs continuously decreases. Other conditions (extended crystallization time) may lead to nearly completely closed crystalsomes, and we will explore different growth conditions in our future study. We did show that when smaller sized linear PEO was used for stepwise crystallization, closed mBBCs were formed. This further supports that the large size of the mBB plays an important role in the absence of completely closed mBBCs in our earlier experiments.

We agree that while PEO was used to close the opening, the crystals are not perfect spheres. This is not surprising since when linear PEO crystallizes on to a mBBC template, the intrinsic shape of PEO single crystal is not spherical, but flat. The purpose of introducing linear PEO to achieve completely closed BCC was for the FRET experiments, which proved the asymmetry of the mBBC lamellae. This manuscript focuses on the discovery of curved/spherical shaped crystals with broken translational symmetry, which has been clearly demonstrated.

We added: “Note that the mBBCs in Figure 1L do not have a perfect spherical shape. This could be because the intrinsic shape of a linear PEO single crystal is not spherical, but flat. Future work will be conducted to further explore the detailed mBB growth habit near the closure of the mBBCs.” on page 9, line 7 from the bottom.

6. The self-seeding step is absolutely critical for the authors to observe the spherical habit. Yet, we, the readers are left with this being a magical step in the process and I consider this to be one of the most important parts of this manuscript that is left completely open. What is the nature of the seed crystal? It is at this stage that the die is cast. In order to comprehend the manuscript fully, a deeper discussion of the seed crystals is necessary.

R: Crystal seeds are used as the heterogeneous nucleation sites for polymer crystallization so that all the crystals start to grow at the same time and uniform crystals can be formed. To answer the reviewer's question, we conducted dynamic light scattering (DLS) experiments. The following figure shows the DLS results on a dissolved mBB sample at 80 °C (a), polymer solution with seeds at 44.5 °C (b) and solution after crystallization at 30 °C (c). In a, the narrow peak indicates the size of molecular bottle brush molecules. In b, in addition to the mBB peak, there is another small peak at 200 nm, corresponding to the crystal seeds. In c, the two peaks can be assigned to dissolved polymer and mBBCs. The molecular bottle brush molecule peak gradually shifted to large size, which may be because as temperature decreases, the polymer slightly aggregated.

Dynamic light scattering of (a) completely dissolved mBB solution; (b) A mBB seed solution before crystallization and (c) mBBC suspension after crystallization

We added this plot in the supporting information, and in the discussion, we added “The crystal seeds that remained at T_{ss} can be confirmed using dynamic light scattering experiments (**Figure S8**). They provide heterogenous nucleation sites for subsequent crystal growth and the self-seeding effect on crystal growth is summarized in **Figure S9**.” On page 5, line 9 from the bottom.

To further prove that the curved morphology is the intrinsic morphology of the mBBC crystals, we used large flat PEO crystal as seeds and curved mBBCs can be grown, as shown in the following figure.

SEM image of mBBC crystals formed using a flat (central part) PEO single crystal as the seed.

7. The authors make arguments about the crowding giving rise to the observed spherical habit. Yet, they also make a statement about the immiscibility of the backbone with spacer and the PEO side chains. Wouldn't this play a critical role in segregating the PEO chains to enable the crystallization to proceed in the manner that it does? Some thermodynamic arguments about the state of miscibility at the formation of the seed crystals is in order. One need to take the PEO of

both sides of the brush, bring them to one side (which will require energy) and then get them crystallize. It is at this stage where the growth begins that templates everything else.

R: We estimated the total free energy of the mBB upon crystallization in two cases, as shown in the following figure:

In case 1, the PEO side chains form crystals on both sides of the backbone while in case 2, the side chains form crystals on one side of the backbone. Assuming the crystal thickness is l and the lateral dimension is a . For simplicity, we assumed the two lateral sides have the same dimension a . σ_1 , σ_2 , σ_3 are the surface free energies of the crystal lateral surface, top surface, and the surface in contact with backbone.

For case 1, we have $\Delta G = 8la\sigma_1 + 2a^2\sigma_2 + 2a^2\sigma_3 - \Delta g$

For case 2, we have $\Delta G = 6la\sigma_1 + 2a^2\sigma_2 + 2a^2\sigma_3 - \Delta g$

ΔG is the free energy of the crystal, and Δg is the bulk free energy of fusion. From the above two equations, we can see that case 2 has a lower free energy by $2la\sigma_1$. The asymmetry nature was also confirmed by the FRET results.

We added this discussion in the supporting information. In the revised manuscript, we added “The asymmetrical nature of the nBBC shell can also be supported by a thermodynamic argument, detailed in the supporting information (**Figure S14**).” On Page 17, line 2.

8. “ where micro-faceted edges (red arrows in Figures 1E) further confirms the crystalline nature “ Doesn’t this say there must be a registry of the crystals and, if this were in 2D, there would be no translational symmetry issues. I might add that, given the density of data put into each figure, the micrographs are bordering on the verge of being too small to discern the edges of the spherical shells where this faceting is evident. A magnification of this would have helped.

R: Yes the facets further confirm registry of the crystal. Even in 2D, the lattice direction continuously changes on curved surfaces. See for example “Grain boundary scars and spherical

crystallography. *Science* 2003, **299**, 1716-1718”. We added an enlarged facet image in Figure 1E to highlight the microfacets.

9. In Figure 1 there is a schematic of the brush polymer that one sees everywhere. That is fine but I would like to have seen a similar schematic for a single chain in the crystals that are formed. Certainly, the side chains must be on one side of the main chain. Along these lines, I have gotten no sense of the behavior of the main chain when the system crystallizes. For the planar single crystals somewhat, linear squiggles are shown in Fig.2C. Do the authors mean to infer that the backbones are stretched out in that manner. For the spherical crystals, even that is not provided. However, the configuration of the backbone chain must be an important consideration, yet it is left untouched. This must also be an important consideration when the authors tried to complete the spheres with the addition of more bottlebrush polymer (that failed). I am really left wanting without any information on the backbone chains and that is certainly critical for the crowding arguments as well.

R: Figure 2D is the schematic representation for mBBC, which is reproduced in the following:

We consider that the main chain conformation would be close to an extended conformation, similar to the original bottle brush (in A) and in a symmetric crystal (B). Figure D shows the enlarged region of (C). As we wrote in the caption of Figure 2: “Two mBB molecules are shown in (D) with two adjacent side chains in each mBB. Each side chain crystallizes into a small yellow rectangular frustum with defined chain folding, while the blue rectangular frustum represents the volume of the closely packed amorphous layer.....” . For clarity, we only showed two PEO side chain frustums for each mBB molecule. A more complete scheme for one mBB molecule can be seen as following. The backbone is close to an extended chain conformation.

We revised Figure 2D:

And the new figure caption reads:

Figure 2. mBBC formation mechanism. An mBB (A) could crystallize following pathway i to form a flat symmetric crystal (B). The blue shaded areas denote mBB backbones/spacers. Pathway ii leads to an asymmetrical crystal (C, D). x and z denote growth direction and the PEO axis, respectively. Two mBB molecules are shown in (D) with two adjacent side chains in each mBB. Note that each PEO side chain forms multi-stem layers (yz plane) orthogonal to the growth direction x to alleviate local packing crowdedness. The local chain crowdedness in high grafting density mBB polymers leads to cross-sectional area mismatch between the mBB backbone and side chain crystals, which eventually induces lamella bending, translational symmetry breaking and the subsequent formation of mBBCs. The inset in (D) highlights one side chain per mBB molecule. The side chain crystallizes into a small yellow rectangular frustum with defined chain folding, while the blue rectangular frustum represents the volume of the closely packed amorphous layer.

10. If I were to consider all the PEO chains in the bottlebrush polymers, what fraction of the chains are incorporated into the crystals?

R: All chains are incorporated into crystal and based on the DSC experiments, the crystallinity is approximately 60%. The segments that are linked to the spacers as well as the folds would be in the amorphous fraction.

11. “The measured thicknesses are 9.7, 10.5 and 12.7 nm, indicating thicker lamellae in larger mBBCs . This also argues that there is no chain folding, since there would be an integral number of stems. However, I ask, what happens to the melting points and does that change systematically with crystal thickness?

R: As discussed in the previous response, PEO side chains do fold in the mBBC crystals, and the fold number can be calculated using equation 5 in the manuscript: $\zeta = (1.939 \times \frac{m}{l_{sc}} - 1)$, where ζ is the fold number, m is the degree of polymerization of PEO side chain, l_{sc} is the

lamellar thickness. The molecular weight of the PEO side chain is 5000 g/mol, PEO degree of polymerization is $5000/44 \approx 114$, and the measure l_{sc} ranges from 9.7-12.7 nm. ζ can therefore be calculated to be ~ 2.3 to 1.5, indicating that each side chain folds 1.5 to 2.3 times. This indicates that non-integral folding occurred in the system, which is not uncommon for 5000 g/mol PEO.

We conducted differential scanning calorimetry experiments on mBBCs with the crystallization temperature (T_c) at 20, 25, 30 degree Celsius. The first heating, first cooling and second heating results are shown in the following figure and table. The melting point from second heating curve can be considered as the melting point of bulk molecular bottle brushes. From the table, we can conclude that when increasing the crystallization temperature, mBBC melting points was similar for $T_c = 20$ and 25 °C, which is slightly lower than that of $T_c = 30$ °C. mBBC melting points are slightly higher than the melting point of bulk mBBCs (~ -0.4 - 0.9 °C). The slight melting points change can be associated with the differences of crystal lamellar thickness and the mBBC diameter at different T_c .

DSC scan of mBBCs crystallized at different temperature. (a) first heating, (b) first cooling, (c) second heating.

Crystallization temperature	First heating melting temperature	Second heating melting temperature
20 °C	56.2 °C	55.8 °C
25 °C	56.1 °C	55.5 °C
30 °C	56.5 °C	55.6 °C

Reviewer #2 (Remarks to the Author):

The manuscript by Christopher Li and co-authors describes a very interesting case of non-planar crystals formed by the bottlebrush polymers with crystallizable side chains. The studied crystals have a very unusual shape of hollow spheres. The authors emphasize the absence of translational symmetry for such objects claiming that this case exemplifies a new principle of spontaneous translational symmetry breaking.

The paper is very interesting and certainly deserves publication. However, several issues have to be addressed before it is accepted:

R: we appreciate the positive comments from the reviewer.

1. The authors would like to provide more details about previous reports on non-planar polymer crystals. In particular, I would propose to be more specific in describing the known crystal habits such as helicoids, scrolls, 3D helices as well as the experimental methods allowing to address the details of these morphologies including synchrotron-based micro- and nano-focus X-ray scattering.

R: we have added discussion on non-planar polymer crystals in the revised manuscript, page 3, line 2.

“Detailed reasons for the formation of these shape-symmetry incommensurate crystals are material-specific, while unbalanced stress is believed to be an important reason for symmetry breaking.⁵ For example, in polyethylene, unbalanced stress can arise from chain tilting with respect to the lamellar normal, leading to banded spherulites comprised of helicoidal crystals.^{5,9-11} Lamellar unbalance can also be induced by different volumes of the folds as proposed in γ phase poly(vinylidene fluoride) (PVDF) and polyamide 66.^{5,8,11,12} Triblock copolymers with crystalline middle block and immiscible end blocks can form asymmetric curved crystals and the unbalanced stress is associated with phase separation of two end blocks.¹³ In addition, chiral structure can also lead to symmetry breaking upon forming single crystals.^{6,7,14}”

2. In relation to p.1, it would be desirable if the authors can propose a structural model explaining the morphogenesis of such spherical-shaped crystals. It will be worth comparing the formation of all main non-planar crystal habits together with the newly discovered hollow spheres.

R: We appreciate the constructive suggestion. As we discussed in the manuscript, the origin of the bending of lamella is from unbalanced layer structure: the mBBC shells have two layers (Crystalline mBB and amorphous spacer/backbone, as shown in Figure 2D). The different nature of the two layers (packing, density etc.) leads to the lamellar bending. This is similar to the reported unbalanced folding of PVDF and Nylon scrolled single crystals.

We revised Figure 2 and added “The asymmetric nature of the mBBC shells is the origin of lamellar bending. Similar observations have been reported in PVDF and polyamide 66 scrolled

single crystals,^{8,12} where the unbalanced folding in these two cases leads to lamellar asymmetry and the subsequent crystal bending. In triblock copolymer polystyrene-*b*-poly(ethylene oxide)-*b*-poly(1-butene oxide) (PS-*b*-PEO-*b*-PBO), upon PEO crystallization, PS and PBO respectively separate into top and bottom surfaces of the PEO single crystal, leading to crystal bending. Note that in all these cases, lamellae are considered as a three-layer structure with a crystalline central layer while the top and bottom layers are either fold surface (PVDF and polyamide 66) or PS/PBO domains. In the present case, we use a two-layer model, which introduces similar asymmetry that can lead to lamellar bending. The asymmetrical nature of the mBBC shell can also be supported by a thermodynamic argument, detailed in the supporting information (Figure S14).” On page 16, line 8 from the bottom.

3. The referee would like to learn more about the mechanics of such objects. Can the authors rationalize the fact that they preserve their 3D shape after deposition on the substrate. What are the conditions for them to keep their 3D shape.

R: We consider an elastic shell putting in contact with a solid surface via a layer of liquid (mimicking the drop casting process), and deformed by δ due to the elastocapillary force,

R is the radius of the shell, a is the contact area diameter. Based on classical shell theory and contact mechanics (*J. Phys. Cond Matt.* **2010**, 22, 493101), free energy gain by increasing the contact area is $\sim \gamma a^2$, where γ is the surface free energy of the liquid. The corresponding elastic energy of the shell is $\sim E a^3 (h/R)^{5/2}$ (Landau, LD, Lifshitz, EM, *Theory of Elasticity*, 3rd Ed., 1986). E is Young's modulus of the shell, h is the shell thickness. $\gamma = 0.012$ N/m (<https://pubchem.ncbi.nlm.nih.gov/compound/Pentyl-acetate#section=>) for pentyl acetate and $E = 3$ GPa (*Macromolecules* 2011, 44, 7758-7766.), $h \sim 10$ nm, and $R \sim 1200$ nm. Balancing two terms we have $a \sim \frac{\gamma}{E} \left(\frac{R}{h}\right)^{5/2} \sim 631$ nm. Since $a < 2R$, and the first term scales with a^2 while the elastic energy loss scales with a^5 , the shell would deform to $a \sim 631$ nm at the bottom, and the top part of the spherical shape will not be significantly distorted.

Note that this is a first order estimate, factors such as the lack of perfect spherical shape, chain sliding and chain orientation during the deformation in the shell could also influence the calculation. It nevertheless provides useful understanding of the mechanical stability of mBBCs upon sample preparation. Detailed mechanical analysis would be of interest and will be pursued in our future work.

Reviewer #3 (Remarks to the Author):

This communication reports an unusual crystallization behavior of bottlebrush polymer with crystallizable side chains. Crystallization-induced self-assembly of PEO brushes creates microspheres with a curvature controlled by grafting density. Lower grafting density would lead to larger 'crystalsomes'. This finding is significant and merits publication.

R: we appreciate the positive comments from the reviewer.

The central hypothesis of this paper is segregation of all side chains on one side of crystalline lamellae. This segregation results in spontaneous curvature and ultimately yields spherical shells. This hypothesis makes sense, however, needs better justification. The crowdedness argument is relevant to conformation of brush molecules in solution. Crystallization morphology is largely controlled by kinetics and short-range interactions. The backbone plays role of a kinetic constraint and a packing defect. Such defects tend to segregate at the surface of crystallites. In sum, the origin and molecular mechanism of curvature is unclear to me.

R: Crowdedness can exist in solution and is retained in bulk. Crystallization in dilute solution for long crystallization time is close to thermodynamic equilibrium. We attributed the origin of the crystal bending to the asymmetric lamella whose top and bottom surfaces are different. This argument is similar to the unbalanced fold surface observed in PVDF and Nylon 66 scrolled single crystals and triblock copolymer system with crystallizable middle block and immiscible end blocks. The side chain arrangement relative to the backbone has been approved by the FRET experiment. The reason for inadequate volume is due to the fixed backbone length which provides a limited space for the side chains to crystallize. This was further proved by showing that the backbone effect in inducing curved crystalline lamellae disappeared when the grafting density of side chains decreased to 0.1. The reviewer also raised another good point and we are interested in studying backbone's kinetic constraint role in our following research.

We added "The asymmetric nature of the mBBC shells is the origin of lamellar bending. Similar observations have been reported in PVDF and polyamide 66 scrolled single crystals,^{8, 12} where the unbalanced folding in these two cases leads to lamellar asymmetry and the subsequent crystal bending. In triblock copolymer polystyrene-*b*-poly(ethylene oxide)-*b*-poly(1-butene oxide) (PS-*b*-PEO-*b*-PBO), upon PEO crystallization, PS and PBO respectively separate into top and bottom surfaces of the PEO single crystal, leading to crystal bending. Note that in all these cases,

lamellae are considered as a three-layer structure with a crystalline central layer while the top and bottom layers are either fold surface (PVDF and polyamide 66) or PS/PBO domains. In the present case, we use a two-layer model, which introduces similar asymmetry that can lead to lamellar bending. The asymmetrical nature of the mBBC shell can also be supported by a thermodynamic argument, detailed in the supporting information (Figure S14).” On page 16, line 8 from the bottom.

Minor comments:

1. The introduction signifies the novelty of this work by comparing it to chiral and bending polymer crystalline domains. However, the two paragraphs can be better streamlined if the author would focus on specific driving forces behind crystallite curvature.

R: we can revise this.

2. Fig 2d is misleading as it suggests each side chain forms its own domain

R: We modified the figure based on the reviewer’s comments. The revised Figure 2D is shown below.

3. The purpose of introducing Eq.6 needs justification. From its current state, d_{bb} can only be calculated from p_c and not from measurement, making this equation unverifiable with data presented in this work. Meanwhile, any changes to the chain architecture or thermal history will also change d_{bb} , making the predictions to self-assembly behavior inaccurate.

R: We agree that d_{bb} cannot be directly measured. We use this to guide our understanding of molecular packing. We added: “Note that d_{bb} can be used to guide our understanding of the mBB packing, and can be significantly influenced by grafting density, side chain length and crystallization temperature.” on page 14, line 5 from the bottom in the revised manuscript.

4. The outcome of tests regarding l_{sc} and σ proves the theory that the backbone and sidechain phase separates and the side chains are on the outside and less dense side chain would result in less curvature. However, the changes are not on a similar scale, contrary to the suggestions of

prediction 2 from eq.6. As an example, when comparing samples with $\sigma=0.48$ vs 0.94 . σ changed by $\sim 100\%$ yet p_c changed by 0.4% . Obviously, d_{bb} and/or l_{sc} also changed. Though the trend predicted is correct, it cannot be deducted quantitatively from eq.6.

R: Equation 6 ($p_c \equiv \frac{a_{bb}}{a_{sc}} = \frac{4.23d_{bb}l_{sc}}{\sigma m}$) was derived based on the volume calculation of the side chain crystals and the amorphous portion of the chain. In the case of $\sigma = 0.48$ and 0.94 , the review is right that grafting density increased by 100% while p_c changed by a much lesser extended. This, as the reviewer pointed out, is due to the change of d_{bb} and/or l_{sc} . We also acknowledge that the prediction is rather qualitative because of the possible variation of d_{bb} and/or l_{sc} .

We added “Note that d_{bb} can be used to guide our understanding of the mBB packing, and can be significantly influenced by grafting density, side chain length and crystallization temperature. While comparing mBBCs formed by polymers with different σ , because d_{bb} and l_{sc} could also change, using equations 6 and 7 provides a qualitative understanding of grafting density dependence of mBBCs (see later results).” On page 14, line 5 from the bottom.

We hope that we have addressed all the comments and that the revised manuscript is acceptable for publication in Nature Communications. Thank you very much for your consideration and I will be happy to provide any further information if needed.

Reviewers' comments:

Reviewer #1 (Remarks to the Author):

I still feel that his manuscript needs some more work prior to publication. See my comments below in response to the authors rebuttal.

1. I am really not satisfied with this response to point 1. There is no question that different orientation of the crystals will give rise to a broadening azimuthally. If there is a splaying, then I would expect that the broadening would occur ONLY toward the higher diffraction vector side, i.e. the broadening would be asymmetric. Is this the case? What the authors provide in this figure is really useless to discern the nature of the broadening. Also, does the broadening occur uniformly for all reflections? Are these broadenings symmetric or asymmetric? From the schematic that the authors show, it certainly cannot be symmetric and should be reflected in the data. X-ray data is not, in my opinion, the relevant data to support the authors arguments and electron diffraction is still needed. They are proposing a new model, in fact original claimed a symmetric breaking. While this has been removed from the title, the fact remains in their arguments and I am not convinced of this splaying argument. While the authors refer to an earlier work, that does not make it correct. I am certain that cryo-TEM could be used to obtain these data. Since smaller crystals are formed giving rise to the broadening, am I to conclude that splaying does not contribute to the broadening? There is a Catch-22 here and a more detail analysis of the diffraction data is needed and I still feel that the cryo-TEM is needed for the authors to support their claims. The packing of the crystals reminds me of arguments that people have made in the storage industry where simple hexagonal or cubic packing is not suitable for data storage on a circular disk (we can make it simpler than the spherical surface the authors are dealing with). Strain induced in the packing of these unit cells forces a resetting of the cell as one proceeds from the center of the disk outward and, also, tangentially around the disk. The problem is the identical one that the authors are dealing with. The storage industry had to resort to sectoring. In the authors case, the crystal size is reduced. This, however, reduces the strain on the lattice significantly. Translational symmetry is recovered from the sectoring, though this requires a reset of the origin. I can't see why something similar would be happening here.

2. Relevant to the discussion in point 1, I would like to know if the translation symmetry breaking is a result of the small crystal size (which would break the translational symmetry) or is this a result of an actual splaying?

3. I am satisfied with the response to point 3.

4. OK

5. Not too gratifying a response but the manuscript is revised.

6. I presume the Flat PEO Seed Crystal image will appear in the supporting information.

7. OK

8. Again, the symmetry breaking is due to the authors remaining with Cartesian coordinates and I really do not feel that is appropriate for the studies here. It seems like the authors want to have their cake and eat it too. It is for me, the reader, confusing and I would be happier with the spherical coordinate use.

9. Response to comment 9 is OK. Schematics can be drawn but believing them is another matter. A simulation would be more gratifying.

10. OK

11. OK

Reviewer #2 (Remarks to the Author):

The referee is satisfied with the authors responses.

Reviewer #3 (Remarks to the Author):

The authors fully and consistently replied to the reviewers comments. Therefore, the paper is recommended for publication as submitted.

Reviewers' comments:

Review 1:

I still feel that his manuscript needs some more work prior to publication. See my comments below in response to the authors rebuttal.

Question 1. I am really not satisfied with this response to point 1. There is no question that different orientation of the crystals will give rise to a broadening azimuthally. If there is a splaying, then I would expect that the broadening would occur ONLY toward the higher diffraction vector side, i.e. the broadening would be asymmetric. Is this the case? What the authors provide in this figure is really useless to discern the nature of the broadening. Also, does the broadening occur uniformly for all reflections? Are these broadenings symmetric or asymmetric? From the schematic that the authors show, it certainly cannot be symmetric and should be reflected in the data. X-ray data is not, in my opinion, the relevant data to support the authors arguments and electron diffraction is still needed. They are proposing a new model, in fact original claimed a symmetric breaking. While this has been removed from the title, the fact remains in their arguments and I am not convinced of this splaying argument. While the authors refer to an earlier work, that does not make it correct. I am certain that cryo-TEM could be used to obtain these data. Since smaller crystals are formed giving rise to the broadening, am I to conclude that splaying does not contribute to the broadening? There is a Catch-22 here and a more detail analysis of the diffraction data is needed and I still feel that the cryo-TEM is needed for the authors to support their claims. The packing of the crystals reminds me of arguments that people have made in the storage industry where simple hexagonal or cubic packing is not suitable for data storage on a circular disk (we can make it simpler than the spherical surface the authors are dealing with). Strain induced in the packing of these unit cells forces a resetting of the cell as one proceeds from the center of the disk outward and, also, tangentially around the disk. The problem is the identical one that the authors are dealing with. The storage industry had to resort to sectoring. In the authors case, the crystal size is reduced. This, however, reduces the strain on the lattice significantly. Translational symmetry is recovered from the sectoring, though this requires a reset of the origin. I can't see why something similar would be happening here.

Response: we are sorry that our responses to the first question were not clear. Herein we include the original question from the first round of review and provide a **concise** answer in addition to our original response. We then will provide more detailed response to the new questions.

The main question is regarding the azimuthal spreading of the electron diffraction pattern and possible q broadening. Regarding the azimuthal spreading we observed in the electron diffraction pattern (Figure 1 in the manuscript), we would like to mention that it is quite common for non-flat (helical, tubular) crystals as mentioned in the manuscript. The reason for the present observed arc-shaped diffraction is mainly because of the non-flat shape of the crystal. In the present case, c axis is perpendicular to the spherical surface and parallel to the radial direction. When crystals are packed in a non-flat spherical surface, unit cell axes changes from one location to another and therefore lattice splaying occurs in all directions. When this type crystal is subjected to electron diffraction experiment, in an ideal case, only very small domains on the spherical surface satisfy the orientation to yield a $[00l]$ zone diffraction pattern observed in our experiment. Because c axis (or the chain direction) is along the radial direction, these domains are likely from the top and bottom part of the crystal. In experiment, these two domains could have different orientations of the a and b axes. More significantly, when preparing a BBC sample with such a thin wall thickness, the bottom part of the spherical crystal deforms, the out-of-plane a, b axes therefore collapse onto TEM grid surface, leading to possible orientation change of the a, b orientation. This would further cause azimuthal spreading of the diffraction. Because of the multiple factors contributing to the azimuthal spreading, it is challenging to predict if the spreading is symmetric or asymmetric in this case. We claim a translational symmetry breaking in our manuscript, **didn't mention either "symmetric" or "asymmetric" breaking in our manuscript.**

Note that the explanation is consistent with the reviewer's comments on small crystallites patched together in spherical shells. "Lattice splaying" and "small crystallites in a spherical shell" can be viewed as similar concept: "a lattice splaying" is the splaying of a small lattice unit (or small crystallites). As the crystallites approach nanometer scale, "splay" is often used in helical and chiral polymer crystals. In any local area, the orientation of the lattice does not change much. The seemingly large arc observed in the diffraction pattern is because of the multiple reasons discussed above. Based on the reviewer's comment we changed:

*"While the mBB crystal morphologies are different from the flat crystals in **Figure 1A**, the SAED in **Figure 1C** confirms that the crystal structure remains the same except that the spotty diffractions observed in the flat crystal become arc-shaped (**Figure 1C**), which is typical for diffractions from non-flat crystals due to the inevitable lattice splay in curved space.^{7, 12, 34, 42-44}"*, to

*"...while the mBB crystal morphologies are different from the flat crystals in **Figure 1A**, the SAED in **Figure 1C** confirms that the crystal structure remains the same except that the spotty diffractions observed in the flat crystal become arc-shaped (**Figure 1C**), which is typical for diffractions from non-flat crystals.^{7, 12, 34, 42-44}". The crystals can also be viewed as packing of small crystallites progressively changing lattice orientation as they form from the seeds. Multiple domains in the crystal could contribute to an SAED pattern, leading to the observed arc-shape.*

When depositing a spherical thin shell crystal on TEM grids, the bottom part of the crystal could deform, further complicate the azimuthal distribution of the diffraction pattern.” On page 5-6.

Regarding q broadening, as our previous response (XRD analysis) shows, the q broadening is quite small and the XRD peaks are symmetric. From our TEM result, radial intensity profile integration shows it's symmetric (Fig R1). XRD results also suggest the peak is symmetric.

Fig. R1 Selected area electron diffraction of a BBC crystal and the corresponding intensity scan along q direction.

Because the azimuthal broadening is due to crystal orientation change, broadening should apply for all the reflections. In our experimental, because only very small (likely bottom/top part of the sphere) contribute to electron diffraction, only the strongest diffraction plane (120) was observed. We anticipate that similar azimuthal spreading should be observed for other diffraction planes should their signals are strong enough to be recorded.

Regardless, in this manuscript, the diffraction pattern was used to show that the spherical capsules are formed by PEO crystals, and the diffraction arc is broad. The significance of our work is the discovery showing that the crystals formed in molecular bottle brushes are spherical with broken translational symmetry and the molecular origin of this observation is side chain overcrowding. From the electron diffraction data, we are only claiming that the crystal lattice changes directions in a spherical surface. This is evident since c axis of the unit cell is perpendicular to the local surface of the sphere and therefore is along the radial direction. The radial direction changes as we move from one location on the sphere to another.

Regarding cryo-TEM, the BBC crystals are a few microns in diameter. To prepare samples thin enough for cryo-TEM diffraction experiment, they will be distorted, and the crystal orientation will be affected. Therefore, cryo-TEM would not be a suitable method for the present system.

Regarding the sectoring strategy, the review is correct that sectoring can be used to restore translation symmetry in a circular disk or any flat 2D crystals. Nevertheless, circular disks are **flat**, so orientational order can be kept in the entire disk, i.e. translational symmetry is not broken in a circular disk. Hollow spherical shell crystals in our case are **non-flat**, c axis is along the radial direction and changes as we move from one location on the sphere surface to another. The translational symmetry is therefore broken. This is why we claim “Breaking translational symmetry via polymer chain overcrowding in molecular bottlebrush crystallization”. If we use the sectoring strategy in spherical crystals, the chain axis would be perpendicular to the surface at one location (e.g. poles) and parallel to the surface at others (e.g. equator), which is not the case in our experiment because in PEO lamellar crystal, c axis is perpendicular to the lamellar surface (lowest free energy state).

2. Relevant to the discussion in point 1, I would like to know if the translation symmetry breaking is a result of the small crystal size (which would break the translational symmetry) or is this a result of an actual splaying?

Response:

The local crowded packing in the molecular bottle brush molecules leads to non-flat, spherical crystals. Because the crystal shape is non-flat, crystal axes have to change directions in the crystal as we move from one location to another (c axis is along radial direction), hence translational symmetry breaking. As we mentioned previously, lattice splaying and small crystallite size can be viewed as similar mechanisms: “a lattice splaying” is the splaying of a small lattice unit (or small crystallites). As the crystallites are approaching nanometer scale, “splay” is often used in helical and chiral polymer crystals. We changed “While the mBB crystal morphologies are different from the flat crystals in **Figure 1A**, the SAED in **Figure 1C** confirms that the crystal structure remains the same except that the spotty diffractions observed in the flat crystal become arc-shaped (**Figure 1C**), which is typical for diffractions from non-flat crystals due to the inevitable lattice splay in curved space.”^{7, 12, 34, 42-44} to

“...while the mBB crystal morphologies are different from the flat crystals in **Figure 1A**, the SAED in **Figure 1C** confirms that the crystal structure remains the same except that the spotty diffractions observed in the flat crystal become arc-shaped (**Figure 1C**), which is typical for diffractions from non-flat crystals.”^{7, 12, 34, 42-44}. The crystals can also be viewed as packing of small crystallites progressively changing lattice orientation as they form from the seeds. Multiple domains in the crystal could contribute to an SAED pattern, leading to the observed arc-shape. When depositing a spherical thin shell crystal on TEM grids, the bottom part of the crystal could deform, further complicate the azimuthal distribution of the diffraction pattern.” On page 5-6.

3. I am satisfied with the response to point 3.

4. OK

5. Not too gratifying a response but the manuscript is revised.
6. I presume the Flat PEO Seed Crystal image will appear in the supporting information.

A sample Seed crystals are shown in the following image.

Fig. R2. TEM images of BBC seeds. The sample was prepared by seeding the BBC solution at 44.5 °C for 11 minutes and immediately drop casting the seed solution on a TEM grid, which was lightly dabbed on a piece of filter paper, followed by blowing dry with nitrogen. Scale bar 100 nm

7. OK

8. Again, the symmetry breaking is due to the authors remaining with Cartesian coordinates and I really do not feel that is appropriate for the studies here. It seems like the authors want to have their cake and eat it too. It is for me, the reader, confusing and I would be happier with the spherical coordinate use.

Response: We are not aware of any work using a spherical coordinate to describe crystal translational symmetry, perhaps because that a crystal unit cell (a, b, c) is defined using a Cartesian coordinate. Based on the author's suggestion, we added: "Note that since the crystals are spherical, we can also view that the translational symmetry is recovered using a spherical coordinate to define the space." On page 10.

9. Response to comment 9 is OK. Schematics can be drawn but believing them is another matter. A simulation would be more gratifying.

Response: Thank you for your suggestion. We are interested in collaborating with other research groups to conduct molecular simulation on the BBCs. We look forward to reporting the results in the future.

10. OK

11. OK

We hope that we have addressed all the comments and that the revised manuscript is acceptable

for publication in Nature Communications. Thank you very much for your consideration and I will be happy to provide any further information if needed.

REVIEWERS' COMMENTS:

Reviewer #1 (Remarks to the Author):

The authors have addressed the comments that I raised in my earlier review. How, I just cannot get excited about this work.